# Defining and measuring multimorbidity in primary care in Singapore: Results of an online Delphi study

Shilpa Tyagi[1], Victoria Koh[1], Gerald Choon-Huat Koh[1]*, Lian Leng Low[2], Eng Sing Lee[1,3,4]

1 MOH Office for Healthcare Transformation (MOHT), Singapore, Singapore, 2 Department of Family Medicine and Continuing Care, Singapore General Hospital, Singapore, Singapore, 3 Lee Kong Chian School of Medicine, Nanyang Technological University, Singapore, Singapore, 4 Clinical Research Unit, National Healthcare Group Polyclinics, Singapore, Singapore

* Gerald_koh@moht.com.sg

**Data Availability Statement:** The datasets generated and/or analysed during the current study are not publicly available because they may contain potentially sensitive information related to the views shared by the Delphi panelists under

## Abstract

Multimorbidity, common in the primary care setting, has diverse implications for both the patient and the healthcare system. However, there is no consensus on the definition of multimorbidity globally. Thus, we aimed to conduct a Delphi study to gain consensus on the definition of multimorbidity, the list and number of chronic conditions used for defining multimorbidity in the Singapore primary care setting. Our Delphi study comprised three rounds of online voting from purposively sampled family physicians in public and private settings. Delphi round 1 included open-ended questions for idea generation. The subsequent two rounds used questions with pre-selected options. Consensus was achieved based on a pre-defined criteria following an iterative process. The response rates for the three rounds were 61.7% (37/60), 86.5% (32/37) and 93.8% (30/32), respectively. Among 40 panellists who responded, 46.0% were 31–40 years old, 64.9% were male and 73.0% were from the public primary healthcare setting. Based on the findings of rounds 1, 2 and 3, consensus on the definition of a chronic condition, multimorbidity and finalised list of chronic conditions were achieved. For a condition to be chronic, it should last for six months or more, be recurrent or persistent, impact patients across multiple domains and require long-term management. The consensus-derived definition of multimorbidity is the presence of three or more chronic conditions from a finalised list of 23 chronic conditions. We anticipate that our findings will inform multimorbidity conceptualisation at the national level, standardise multimorbidity measurement in primary care and facilitate resource allocation for patients with multimorbidity.

## Introduction

Multimorbidity is defined as the presence of multiple chronic health conditions in a single individual [1]. It is common in the primary care setting [2, 3] and has serious implications for both the patient [4–6] and the healthcare system [7, 8]. While it is essential to invest resources and develop interventions to address multimorbidity, there remains ambiguity in defining and

anonymity. Moreover, prior consent was not sought from Delphi panelists to share the data containing their responses outside of the study team. The datasets used and analysed during the current study are available on reasonable request from the following source: - Name: Dr. Praveen Deorani, Data Science & Technology (DST) Unit - Organization: MOH Office for Healthcare Transformation (MOHT), Singapore - Email: praveen.deorani@moht.com.sg.

**Funding:** The author(s) (GCK and LES) received funding from the MOH Office for Healthcare Transformation (MOHT), Singapore (https://www.moht.com.sg/) for this study. The funders had no role in study design, data collection and analysis, decision to publish, or preparation of manuscript.

**Competing interests:** The authors have declared that no competing interests exist.

measuring multimorbidity [9–11]. An earlier review identified multiple definitions of multimorbidity without much convergence [12]. While the World Health Organisation defines multimorbidity as "*being affected by two or more chronic health conditions*" [13], and the Agency for Healthcare Research and Quality defines multiple chronic conditions as presence of "two or more chronic physical or mental health conditions" [14], the European General Practice Research Network adopts a more comprehensive approach and defines multimorbidity as "*any combination of chronic disease with at least one other disease (acute or chronic) or bio-psychosocial factor (associated or not) or somatic risk factor*" [14, 15]. While the definition by World Health Organisation is limited to chronic conditions, the definition by European General Practice Research Network includes both chronic and acute conditions, along with risk factors and biopsychosocial factors. None of the above recommend a list of conditions to be used in defining multimorbidity, which may result in different prevalence estimates generated across different settings using the same definition.

There is no universal framework for adoption locally for defining multimorbidity with several areas of heterogeneity, namely, 1) definition of chronic conditions, 2) the list of conditions considered, 3) the cut-off for the conditions used, and 4) the data sources to confirm conditions. There is reported lack of consistency in defining a chronic condition [10, 16], considering it has been defined as a multi-faceted concept [16]. Two systematic reviews have reported the list of conditions to vary between 5 to 335 [17], and from 2 to more than 50 [18], highlighting heterogeneity in the reported list of conditions. Three systematic reviews reported varying cut-offs (2, 3 or 4 or more conditions) used by included studies for defining multimorbidity illustrating the heterogeneity and lack of consensus in adoption of a cut-off for the number of conditions [10, 17, 18]. Data sources to measure multimorbidity were also reported to vary between patient self-reports (55% of studies) or medical records and administrative databases (42% of studies) or a combination of the two (1% of studies) further illustrating the heterogeneity [18]. Depending on how multimorbidity is measured, the prevalence can range from 13% to 72% in population-based studies and 4% to 99% in primary care-based studies [10]. This variation is concerning as it impedes the quantification of multimorbidity burden and subsequent practice changes to improve patient outcomes. The existing areas of heterogeneity articulated above hold true for Singapore as well. In fact, within Singapore, one study reported that the prevalence of multimorbidity in primary care varied between 6% to 17% [19].

It is important to describe multimorbidity within a specified setting as its prevalence has been reported to vary across different settings [10], with reliability of measures in one setting not translating completely into another setting [20]. Moreover, previous studies have also adopted such a setting-specific approach [17, 20]. We focussed on primary care specific definition of multimorbidity as the commonly seen conditions in primary care (hypertension, depression, anxiety, arthritis etc.) [21] differ from other care settings like emergency department (injuries, lower respiratory tract infections, poisonings etc.) [22] or inpatient setting (septicaemia, pneumonia, complications of diabetes etc.) [23, 24], necessitating a relatively different list of conditions to measure multimorbidity. With this setting-specific approach, primary care is aptly suited to provide patient-centred holistic care for patients with multiple chronic conditions based on the tenets of continuity, coordination and comprehensiveness [25]. Almost eight in ten consultations in a primary care setting may involve a patient with multiple chronic conditions [26]. From a management perspective, primary care setting is aptly suited to manage multimorbidity with the ability to adopt the paradigm of "goal orientation" as opposed to "disease orientation" [27]. Additionally, it is established that multimorbidity management within primary care setting is more cost-effective as compared to other care settings [28]. Hence, for the current study, we focussed on developing a consensus-derived definition of multimorbidity within the primary care setting.

With current lack of a universally recommended approach to defining multimorbidity, it is important to adapt the existing multimorbidity concepts based on contextual relevance and consensus opinion of experts since "locally-generated research evidence" is more likely to be translated into practice as it is highly valued by the policy makers and practitioners [29]. Moreover, the burden of commonly seen conditions is reported to vary by different regions [21]. Hence, the current study is based within Singapore's primary care setting, which is also aligned with the intention of translating the research findings from this study into practice changes to improve multimorbidity management.

From an implementation perspective, it is essential to engage local stakeholders as the research efforts aimed at improving outcomes of intended end-users at scale have the greatest impact when the knowledge generation and application are shared between researchers and stakeholders [30, 31]. This shared responsibility is achieved by co-production of knowledge between researchers and relevant stakeholders [29]. The potential implication of developing this consensus-derived definition of multimorbidity will be providing a standardised approach to measuring multimorbidity, which will serve as a pre-requisite for planning, resource allocation and programme implementation to improve outcomes of patients with multimorbidity at the national level. This will eventually support development of relevant clinical practice recommendations to help clinicians better manage such patients, ensure continuity of care [32], improve patient outcomes like quality of life [33] and prevent adverse outcomes related to polypharmacy [34]. While local efforts within Singapore to describe multimorbidity are promising [5, 8, 35–37], there still remains heterogeneity in both the overall methodology and the list of chronic conditions considered. Thus, we aimed to conduct a Delphi study to gain consensus on the definition, conceptualisation of multimorbidity, and the list of chronic conditions used to measure multimorbidity in Singapore's primary care setting.

## Methods

### Design

Consensus designs like the Delphi technique or the Nominal Group Technique are "*group facilitation approaches which aim to determine the level of consensus among a group of experts (stakeholders) by aggregation of opinions into refined agreed opinion*" [38]. The consensus-based approach was chosen as it allows balanced participation from all group members compared to qualitative approaches like focus group discussions which may be dominated or influenced by individual group members. We specifically used the Delphi technique for consensus building in our study considering the time and geographical challenges of scheduling face-to-face discussions under the Nominal Group Technique [39]. Additionally, our study was conducted during the COVID-19 pandemic, which further limited mobility and social gathering. Our study was approved by the National University of Singapore's Institutional Review Board (NUS-IRB-2020-248). Findings are reported in accordance with the Recommendations for the Conducting and REporting of DElphi Studies (CREDES) [40] (Please refer to **S1 Appendix**).

### Consensus criteria

For Likert scale rating questions rated from 1 (labelled "of limited importance for making a decision") to 9 (labelled as "critical for making a decision"), the scores were categorised into the following three categories: 1–3 (limited importance), 4–6 (important but not critical) and 7–9 (critical) [41]. A pre-defined standardised threshold for consensus was applied for quantitative Delphi rounds (2 and above): any item with a rating of 7–9 by 70% or more of the panellists and 1–3 by 15% or fewer panellists was included in the conceptualisation of multimorbidity. Any item with a rating of 1–3 by 70% of the panellists and 7–9 by 15% or

fewer panellists was excluded. All other combinations indicated indeterminate or no consensus response and were carried forward to subsequent Delphi rounds till the above consensus criteria were met [41–43]. For categorical questions with responses as *yes/no*, any item with 70% or more of the panellists responding *yes* was included in the conceptualisation of multimorbidity. Any item with 70% or more of the panellists responding *no* was excluded. All other combinations indicated indeterminate or no consensus response and were carried forward to subsequent Delphi rounds till the above consensus criteria were met. For a question with different sub-components requiring *yes/no* responses, if consensus was reached on 80% of the sub-components (e.g., 5 out of 6 sub-components), then consensus at the question level was achieved. If not, the sub-components without consensus were carried forward to the subsequent Delphi round.

## Expert panel members

For the current study, an expert in multimorbidity was defined as a family physician practising family medicine in the ambulatory primary care setting in Singapore who regularly encountered patients with multiple chronic conditions in either public or private settings. Additionally, a person was considered an expert in multimorbidity if they were recognised as an expert by their peers (e.g., recommended as those with relevant expertise to contribute towards the Delphi panel). Participants were excluded if (1) they practised in non-ambulatory primary care setting like community hospitals in full or partial capacity or (2) were unable to commit for the entire Delphi process. Since the recommended sample size for Delphi panels ranges between 10 to 50 [44], we aimed to recruit a total of 50 panellists. Factoring in a response rate of about 70%, we sought 70 nominations.

## Recruitment of expert panel members

Adopting purposive sampling and leveraging on the study team's contacts, we engaged key personnel in each of the following five institutions/organisations: 1) National Healthcare Group Polyclinics, 2) National University Polyclinics, 3) SingHealth Polyclinics, 4) Singapore Medical Association and 5) College of Family Physicians, Singapore. These key personnel nominated family physicians who met the eligibility criteria. The selection of these institutions/organisations as recruitment sources was based on getting a comprehensive representation of family medicine practitioners from both public and private primary care settings within Singapore. Potential panellists were invited to participate in the Delphi study via email, describing the purpose of the study and related details. Additionally, they were informed that agreeing to participate and clicking on the link to the online survey in the invite email would be taken as them giving consent (i.e., they understood the study details and were willing to participate).

## Delphi rounds

Our Delphi study comprised of three online rounds with round 1 being qualitative enquiry in nature and subsequent two rounds being quantitative enquiry in nature. The core features of anonymity, iteration and controlled feedback, statistical group response and expert input were observed. The SurveyMonkey platform was used for implementing online surveys [45]. Surveys for each round were pilot tested to assess the relevance and readability of questions and the time taken to complete. Necessary amendments were made to the survey before sending it to the panellists. To ensure a high response rate for each Delphi round, research staff sent regular reminders (up to two) to the panellists after the invitation email before labelling them as "no response". These panellists were excluded from subsequent Delphi rounds. Additionally,

we adopted a '*pseudo-anonymity*' based approach whereby respondents were known to only one research team member who coordinated the email invites and reminders, but their judgements and opinions remained strictly anonymous to the rest of the research team and the Delphi panellists [46]. Each Delphi cycle (including survey implementation and analysis of responses) lasted for about four to six weeks. After each round, the panellists were provided a summary of the findings of that round along with their individual responses to inform their participation in the subsequent Delphi round.

Round 1 focussed on idea generation and identification of salient features via asking open-ended questions on describing multimorbidity and gathering views on the preliminary list of chronic conditions (for defining multimorbidity) developed by the research team [19, 47]. (Please refer to **S2 Appendix**) This preliminary list of chronic conditions used as a starting point in the current study was conceptualized based on the originally proposed list of 20 chronic conditions by Fortin and colleagues [47]. This list of 20 conditions was derived from a scoping review of 44 publications with a total of 131 short-listed conditions. The short-listed conditions comprised of a diverse combination of diseases (e.g., myocardial infarction), risk factors (e.g., hypertension), symptoms (e.g., faints) and categories of conditions (e.g., respiratory problems). The proposed list of 20 conditions was derived from the 131 short-listed conditions based on the following selection criteria: relevance in primary care setting, impact on the patient, frequency of reporting of conditions in existing studies and prevalence in primary care patients. The grouping together of certain conditions like angina, myocardial infarction, atrial fibrillation etc. under the category of 'cardiovascular disease' was based on such conditions affecting the same body system. The detailed methodology is described elsewhere [47]. The list proposed by Fortin et al. [47] was modified to make it suitable for use in primary care setting in Singapore. To elaborate, the authors suggested a modified list of conditions for measuring multimorbidity on the basis of relevance in primary care in Singapore, conditions with high burden and number of chronic conditions in a list. The relevance in primary care setting of Singapore was determined by "*consensus reached after iterative discussions between clinicians, research team members and reference to statistics from the MOH and local primary care initiatives*" [19]. A condition was considered to have high burden if the standardised prevalence ratio was 1% and above in the primary care setting in Singapore. The authors considered lists comprising of 12 conditions or more as per the previously recommended appropriate threshold [10]. Additionally, 'pre-diabetes' and 'physical disability' were added to the list to increase the comprehensiveness. Pre-diabetes was added to align with the local context as the Singapore government has placed significant emphasis on management of individuals with diabetes and pre-diabetes [48]. Additionally, pre-diabetes is part of the Chronic Disease Management Program (CDMP) in Singapore, which was introduced in Singapore in 2006 to provide guidance on management of commonly occurring chronic conditions in primary care and reducing out-of-pocket payment for such conditions by introducing financial measures [49]. Physical disability was included as it comprised of conditions like hearing loss which are relevant in the Singapore context due to aging population as well as considering the multi-dimensional impact of such conditions on the patients [19]. While this modified list of conditions was a good starting point for seeking feedback from the Delphi panellists, it is important to note that it could not be directly adopted for measuring multimorbidity since it was recommended based on relative performance in comparison with the other locally used definitions of multimorbidity in Singapore. Secondly, the validity of these definitions could not be assessed completely in the absence of sufficient methodological details. Thirdly, contextual relevance and acceptance by local stakeholders were not known [19]. Lastly, Singapore has a hybrid healthcare system comprising of both public and private providers [50]. Within the public primary care, the primary care clinics are organised into three clusters to better meet

the needs of the patients. Since the above study was based in one of these three public primary care clusters with no representation from private providers of primary care, its findings may not be reflective of the whole primary care in Singapore. Hence, there was a need to seek feedback from stakeholders across both public (including all three clusters) and private settings to derive a consensus-based definition of multimorbidity. Throughout the iterative process of deriving the finalised list of conditions for defining multimorbidity, the total conditions included in the list at all times was more than 12, which is the recommended value for preventing underestimation of multimorbidity prevalence [10].

The data was analysed using content analysis [51]. Based on findings of round 1, the questionnaire for round 2 was adopted. To familiarise the panellists with multimorbidity literature, we shared a slide deck with background information on multimorbidity with the invitation email for round 2. This was done to ensure we obtained unbiased and new ideas from panellists in round 1. For this quantitative round 2 survey, the panellists were asked to indicate their preference by voting on a Likert scale or indicating *yes/no*. Data were analysed using descriptive statistics to determine whether consensus was achieved or not according to the pre-determined criteria. Only topics which did not reach voting consensus were taken to round 3, which focussed on the evaluation of responses from round 2 and consensus-building on the pending topics. The gathering of responses by panellists and subsequent analysis in round 3 were similar to that done for round 2 (Please refer to **S3** and **S4 Appendices** for round 2 and 3 survey questions).

## Results

The response rates for Delphi rounds 1, 2 and 3 were 61.7%, 86.5% and 93.8% respectively. Please refer to **Fig 1** for the study flowchart. Baseline participant characteristics are presented in **Table 1**. Forty-six percent of the panellists were between 31 to 40 years old, 64.9% were male, 51.4% of panellists had Fellowship in Family Medicine [52] and 73.0% practised in public primary healthcare setting.

### 1. Delphi round 1

**1.1. Definition and conceptualisation of multimorbidity.** *1.1.1. Definition of a chronic condition.* The most common elements of the definition of a chronic condition shared by the panellists were whether the condition required long-term management (N = 20), whether the condition was incurable (N = 18) and the duration of the condition (N = 16). Other elements of the definition of a chronic condition reported were impact on the patient (N = 12), persistence or recurrence (N = 8) and associated sequelae, including complications (N = 7) etc.

*1.1.2. Definition of multimorbidity.* About 60% of the panellists reported being aware of a definition of multimorbidity. Most panellists (N = 19) described multimorbidity as the co-occurrence of multiple chronic conditions in a particular individual. Another dimension of multimorbidity definition commonly cited was the cut-off for number of conditions (N = 19); however, the cut-offs provided by panellists varied (e.g., two or more, three or more, more than one etc). Some less commonly reported dimensions of multimorbidity definition included the adoption of a holistic biopsychosocial perspective (N = 4), the inclusion of impact on patient outcomes (N = 5) and the interactive nature of co-existing conditions (N = 5).

Only 21% of the panellists agreed that counts of chronic conditions are sufficient to identify patients with multimorbidity. Some of the reasons included ease of defining multimorbidity with counts, sufficient for research etc. The majority felt that counts are not sufficient for identifying patients with multimorbidity as other considerations should be included, such as complex interaction between different conditions, biopsychosocial perspective, impact on the patient, the severity of each condition etc.

### Delphi Round 1

- Invites Sent, N=60

- Responses Received, N = 40

    o   2 panellists submitted responses twice

    o   1 panellist did not include participant ID

- Responses Analysed, N = 37 (Response Rate = 61.7%)

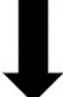

### Delphi Round 2

- Invites Sent, N=37

- Responses Received, N = 32

- Responses Analysed, N = 32 (Response Rate = 86.5%)

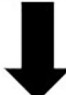

### Delphi Round 3

- Invites Sent, N=32

- Responses Received, N = 30

- Responses Analysed, N = 30 (Response Rate = 93.8%)

**Fig 1. Study flowchart.**

The most commonly reported cut-off was three conditions and above, which was reported by 13 of the 34 panellists. The second most commonly reported cut-off was two conditions and above (8 out of 34 panellists). Other cut-offs suggested by panellists included five or more conditions (5 out of 34 panellists) and four conditions and above (3 out of 34 panellists).

*1.1.3. Measuring the burden of multimorbidity.* Panellists suggested burden of multimorbidity be measured across multiple perspectives (e.g., patient, provider, health system or a combination of these). Patient perspective measures included objective (e.g., disease progression, complications, pill burden) and subjective indicators (e.g., quality of life, mental health scores, etc.). Some examples of provider perspective measures were consultation duration, patient complexity etc. Health system perspective measures mainly included healthcare utilisation and associated cost indicators. The most common response was that multimorbidity should be measured from all three perspectives (i.e., patient, provider and health system). Panellists suggested different sources of data to measure multimorbidity in the general population,

**Table 1. Baseline descriptive characteristics of Delphi panellists.**

| Characteristics of Delphi Panellists | | Number (%) |
|---|---|---|
| **Age** | 21–30 years | 1 (2.7) |
| | 31–40 years | 17 (46.0) |
| | 41–50 years | 11 (29.7) |
| | 51–60 years | 7 (18.9) |
| | More than 60 years | 1 (2.7) |
| **Gender** | Male | 24 (64.9) |
| | Female | 13 (35.1) |
| **Highest Qualification** | FCFP | 19 (51.4) |
| | Mmed (FM) | 15 (40.5) |
| | GDFM | 3 (8.1) |
| **Practice Setting** | Public Primary Healthcare | 27 (73.0) |
| | Private Primary Healthcare | 10 (27.0) |
| **Public Primary Healthcare** | NHGP | 10 (37.0) |
| | NUP | 10 (37.0) |
| | SHP | 7 (25.9) |
| **Part of a PCN[a]** | Yes | 7 (70.0) |
| | No | 3 (30.0) |

**Abbreviations**: FCFP: Fellowship of the College of Family Physicians Singapore; Mmed (FM): Master of Medicine in Family Medicine; GDFM: Graduate Diploma in Family Medicine; NHGP: National Healthcare Group Polyclinics; NUP: National University Polyclinics; SHP: SingHealth Polyclinics, PCN: Primary Care Network.

[a]: only applicable to Private General Practitioners who practice in private primary healthcare setting

including claims data, Electronic Health Records, healthcare utilisation data, medications, absence days, self-reported data and data on social determinants.

**1.2. List of chronic conditions used to define multimorbidity.** *1.2.1. Appropriateness and comprehensiveness of list of conditions*. Panellists shared their views on the appropriateness of the list of chronic conditions developed by the research team to identify people with multimorbidity. Only six out of 34 panellists (who responded to this question) agreed that the list was appropriate and comprehensive enough to be used in the primary care setting. Another six panellists provided general comments on the limited scope of the list.

Apart from depression and anxiety, other mental health conditions suggested by the panellists included schizophrenia, bipolar disorder, childhood mental health disorders like attention deficit hyperactivity disorder, autism, obsessive-compulsive disorder, etc. to expand the mental health conditions included in the list. Apart from hearing loss and congenital malformations, panellists gave various accounts of other conditions which should be included under physical disability, with the two most common suggestions being vision loss/impairment/blindness and loss of limb. Other physical disabilities suggested were neurological (e.g., paraplegia, hemiplegia, etc), severe joint disorders (e.g., gout with tophi, osteoarthritis with permanent deformity etc), soft tissue-related disorders (e.g., tendonitis), chronic paediatric conditions, cognition-related and so forth. Therefore, the original category of '*physical disability*' was re-categorised into the following two categories based on Delphi round 1 feedback: '*functional limitation*' and '*cognitive limitation*'. Inflammatory Bowel Disease (IBD) was not included under the category of 'colon problems' as not many panellists suggested this inclusion. IBD is rare in the Singapore population [53], and patients with IBD are generally seen in specialist setting instead of the primary care setting within Singapore.

The majority (27 out of 34) of the panellists supported the inclusion of chronic pain in the list of conditions and most of them expressed their views from a patient-centred perspective. In qualitative feedback, the panellists shared that chronic pain contributes to morbidity, impacts patient's quality of life, affects physical function as well as mental health and limits socio-occupational functioning. Additionally, long standing pain requires the patient to modify his/her lifestyle and may need regular review and long-term treatment. Panellists who did not support the inclusion of chronic pain in the list viewed it as "*a result of multimorbidity and not multimorbidity itself*" or a symptom of chronic conditions already captured in the list or coded in the system separately. A few panellists acknowledged the challenges associated with capturing chronic pain. Along the same line, some panellists shared that though capturing chronic pain is important, it is difficult to obtain a uniform diagnostic definition or codes on the ground. Based on these recommendations from Delphi round 1, the original list was expanded to include 27 conditions, and this was used for further voting in Delphi round 2 and 3 (Please refer to **S3** and **S4** **Appendices**).

## 2. Delphi round 2

**2.1. Definition and conceptualisation of multimorbidity.** *2.1.1. Definition of a chronic condition.* Consensus was achieved for four (i.e., duration of condition, impact on patient, management of patient and recurrent or persistent course of condition) of the six parameters included in the definition of a chronic condition. The remaining two parameters were moved to Delphi round 3 for further deliberation by panellists (**Table 2**).

For the duration of a chronic condition, only 50% of the panellists voted for six months or more which was closely followed by three months or more (25%). Since none of the options achieved consensus, this parameter was moved to round 3 for further deliberation. More than 70% of the panellists voted for impact on patient to be considered in the following three sub-components: limitations in activities of daily living or instrumental activities of daily living or physical disability (96.9%), mortality (75.0%) and psychological impairment (87.5%). With overall consensus criteria for this question being met, the impact on patient would include the above three sub-components in defining multimorbidity in our context. Further details of voting for parameters of the impact on the patient are provided in **Table 3**.

*2.1.2. Definition of multimorbidity.* The majority of the panellists (84.4%) agreed to the cut-off of three or more conditions to define multimorbidity. Since consensus (68.8%) was not achieved for the sufficiency of counts of chronic conditions for defining multimorbidity, this parameter was moved to round 3 for further deliberation (Please refer to **Table 4**).

*2.1.3. Measuring the burden of multimorbidity.* About 93.8% of the panellists voted for Patient Self-Reported Outcomes as the most appropriate data source for measuring

**Table 2. Findings for parameters for defining a chronic condition.**

| Parameters | Delphi Round 2 | | | Delphi Round 3 | | |
|---|---|---|---|---|---|---|
| | Yes | | Consensus Status | Yes | | Consensus Status |
| | N | % | | N | % | |
| Duration of condition | 23 | 71.9 | Consensus | | | |
| Impact on patient | 26 | 81.3 | Consensus | | | |
| Incurable condition | 21 | 65.6 | No Consensus | 14 | 46.7 | No consensus |
| Management of patient (e.g., long-term follow-up) | 26 | 81.3 | Consensus | | | |
| Recurrent or persistent course of condition | 31 | 96.9 | Consensus | | | |
| Sequelae of condition | 20 | 62.5 | No Consensus | 18 | 60.0 | No consensus |

Table 3. Findings for parameters of 'Impact on Patient' for defining a chronic condition.

| Impact on Patient | Delphi Round 2 | | | Delphi Round 3 | | |
|---|---|---|---|---|---|---|
| | Yes | | Consensus Status | Yes | | Consensus Status |
| | N | % | | N | % | |
| Limitations in activities of daily living or instrumental activities of daily living or physical disability | 31 | 96.9 | Consensus | Not Considered | | |
| Mortality | 24 | 75.0 | Consensus | Not Considered | | |
| Psychological impairment | 28 | 87.5 | Consensus | Not Considered | | |
| Social Deprivation (defined as limited access to society's resources due to poverty, discrimination, or other disadvantage). | 14 | 43.7 | No Consensus | Not Considered | | |

Though consensus was not achieved for the sub-component of social deprivation (43.7%), overall consensus criteria for this question was met (at least 70% of panellists voted yes on at least three of the four sub-components). Hence, the impact on patient would include the former three sub-components in defining multimorbidity in our context and voting for impact on patient was not considered for Delphi Round 3.

multimorbidity from patients' perspective. All 32 panellists voted for Electronic Health Records as the most appropriate data source for measuring multimorbidity from providers' perspective. About 87.5% of the panellists voted for Ministry of Health administrative data as the most appropriate data source for measuring multimorbidity from health systems' perspective. Ministry of Health administrative data was defined as a comprehensive database maintained by the Ministry of Health in Singapore that comprised of health services utilisation data, claims data, and patient medical records.

## 2.2. List of chronic conditions used to define multimorbidity

*2.2.1. Appropriateness and comprehensiveness of list of conditions.* Based on Delphi round 2 findings, 17 conditions out of the list of 27 presented gained voting consensus to be included in the finalised list. The remaining ten conditions were carried to round 3 for further deliberation (Please refer to **Table 5**).

## 3. Delphi round 3

**3.1. Definition and conceptualisation of multimorbidity.** *3.1.1. Definition of a chronic condition.* Under definition of a chronic condition, panellists were invited in round 3 to vote on the two components (i.e., 'incurable condition' and 'sequelae of condition') which did not gain consensus in round 2. Consensus criteria was not met in round 3 as well. For the duration of a chronic condition, the highest proportion of panellists (80%) voted for six months or more, which was the consensus based finalised duration. Therefore, the consensus derived criteria for a condition to be defined as chronic based on finalised four components is as follows: 1) it should last for six months or more (duration); 2) it should be recurrent or have a persistent course (recurrent or persistent course); 3) it should impact the patient in one or more of the following dimensions: limitations in activities of daily living or instrumental activities of

Table 4. Findings for parameters for defining multimorbidity.

| Parameters | Delphi Round 2 | | | Delphi Round 3 | | |
|---|---|---|---|---|---|---|
| | Yes | | Consensus Status | Yes | | Consensus Status |
| | N | % | | N | % | |
| Cut-off of three or more chronic conditions for defining multimorbidity | 27 | 84.4 | Consensus | | | |
| Counts of chronic conditions should suffice | 22 | 68.8 | No Consensus | 26 | 86.7% | Consensus |

**Table 5. Findings for list of chronic conditions for defining multimorbidity.**

| Chronic Condition | Delphi Round 2 | | | | | | | Delphi Round 3 | | | Included in final list |
| --- | --- | --- | --- | --- | --- | --- | --- | --- | --- | --- | --- |
| | Limited Importance (1–3) | | Important not critical (4–6) | | Critical (7–9) | | Consensus status | Include in final list | | Consensus status | |
| | N | % | N | % | N | % | | N | % | | |
| Allergic rhinitis | 14 | 45.2 | 12 | 38.7 | 5 | 16.1 | No Consensus | 6 | 20.0 | No Consensus | × |
| Any cancer in the last 5 years | 0 | 0.0 | 7 | 22.6 | 24 | 77.4 | Consensus | | | | Π |
| Arthritis &/or rheumatoid arthritis | 0 | 0.0 | 6 | 19.3 | 25 | 80.7 | Consensus | | | | Π |
| Asthma, COPD, or chronic bronchitis | 0 | 0.0 | 1 | 3.2 | 30 | 96.8 | Consensus | | | | Π |
| Cardiovascular disease (angina, MI, AF, poor circulation of lower limbs) | 0 | 0.0 | 2 | 6.5 | 29 | 93.5 | Consensus | | | | Π |
| Chronic hepatitis | 2 | 6.4 | 14 | 45.2 | 15 | 48.4 | No Consensus | 26 | 86.7 | Consensus | Π |
| Chronic pain | 1 | 3.2 | 11 | 35.5 | 19 | 61.3 | No Consensus | 27 | 90.0 | Consensus | Π |
| Chronic urinary problem | 0 | 0.0 | 19 | 61.3 | 12 | 38.7 | No Consensus | 23 | 76.7 | Consensus | Π |
| Cognitive Limitation | 0 | 0.0 | 4 | 12.9 | 27 | 87.1 | Consensus | | | | Π |
| Colon problem (irritable bowel) | 1 | 3.2 | 18 | 58.1 | 12 | 38.7 | No Consensus | 16 | 53.3 | No Consensus | × |
| Depression or anxiety | 0 | 0.0 | 2 | 6.5 | 29 | 93.5 | Consensus | | | | Π |
| Dementia or Alzheimer's disease | 0 | 0.0 | 1 | 3.2 | 30 | 96.8 | Consensus | | | | Π |
| Diabetes (including pre-diabetes) | 0 | 0.0 | 0 | 0.0 | 31 | 100.0 | Consensus | | | | Π |
| Functional Limitation | 0 | 0.0 | 8 | 25.8 | 23 | 74.2 | Consensus | | | | Π |
| Gout | 3 | 9.7 | 12 | 38.7 | 16 | 51.6 | No Consensus | 27 | 90.0 | Consensus | Π |
| Heart failure (including valve problems or replacement) | 0 | 0.0 | 2 | 6.5 | 29 | 93.5 | Consensus | | | | Π |
| Hyperlipidaemia | 4 | 12.9 | 5 | 16.1 | 22 | 71.0 | Consensus | | | | Π |
| Hypertension (high blood pressure) | 1 | 3.2 | 3 | 9.7 | 27 | 87.1 | Consensus | | | | Π |
| Kidney disease or failure | 0 | 0.0 | 1 | 3.2 | 30 | 96.8 | Consensus | | | | Π |
| Neurological disorders | 0 | 0.0 | 4 | 12.9 | 27 | 87.1 | Consensus | | | | Π |
| Obesity | 1 | 3.2 | 9 | 29.0 | 21 | 67.8 | No Consensus | 27 | 90.0 | Consensus | Π |
| Osteoporosis | 1 | 3.2 | 8 | 25.8 | 22 | 71.0 | Consensus | | | | Π |
| Other mental health conditions | 0 | 0.0 | 5 | 16.1 | 26 | 83.9 | Consensus | | | | Π |
| Skin conditions | 3 | 9.7 | 14 | 45.2 | 14 | 45.1 | No Consensus | 20 | 66.7 | No Consensus | × |
| Stomach problem (reflux, heartburn, or gastric ulcer) | 7 | 22.6 | 17 | 54.8 | 7 | 22.6 | No Consensus | 13 | 43.3 | No Consensus | × |
| Stroke and TIA | 0 | 0.0 | 1 | 3.2 | 30 | 96.8 | Consensus | | | | Π |
| Thyroid disorder | 0 | 0.0 | 16 | 51.6 | 15 | 48.4 | No Consensus | 28 | 93.3 | Consensus | Π |

**Abbreviations:** COPD: chronic obstructive lung disease, AF: atrial fibrillation, MI: myocardial infarction, TIA: transient ischemic attack

daily living or physical disability, mortality and psychological impairment (impact on the patient) and 4) it should require long-term follow-up (management of patient) (Please refer to **Table 2**).

*3.1.2. Definition of multimorbidity.* With 87% of panellists voting in favour of the sufficiency of counts for defining multimorbidity in round 3, the consensus derived definition of multimorbidity is the presence of three or more chronic conditions from the finalised list of 23 conditions described below (Please refer to **Table 4**).

*3.1.3. Measuring the burden of multimorbidity.* With 96.7% voting for patient Self-Reported Outcomes as the most appropriate data source for measuring multimorbidity from patients' perspective, consensus was achieved for this parameter. With 93.3% voting for Electronic Health Records as the most appropriate data source for measuring multimorbidity from the

providers' perspective, consensus was achieved for this parameter. With 100% voting in favour of MOH administrative data, it would be the recommended data source for reporting multimorbidity from the health systems' perspective.

**3.2. List of chronic conditions used to define multimorbidity.** *3.2.1. Appropriateness and comprehensiveness of list of conditions.* Additional seven conditions received consensus voting in round 3 resulting in the finalised consensus derived list of 23 conditions for defining multimorbidity. (Please refer to **Table 5** for further details and **S5 Appendix** for the finalised list). The consolidated findings for list of conditions across Delphi rounds 1 to 3 are presented in **Table 6**.

## Discussion

With the aim to conduct a Delphi study to gain consensus on the definition, conceptualisation of multimorbidity and the list of chronic conditions used to categorise patients with multimorbidity, we reported the consensus-derived definition as the presence of three or more 'chronic conditions' from the finalised list of 23 conditions, where a '*chronic condition*' is determined based on the following four parameters: 1) lasting for six months or more; 2) be recurrent or have a persistent course; 3) impact the patient in one or more of the following dimensions: limitations in activities of daily living or instrumental activities of daily living or physical disability, mortality and psychological impairment; and 4) require long-term follow-up. This recommended definition will potentially serve as a standardised approach to measuring multimorbidity in primary care setting in Singapore, and enable planning, resource allocation and programme implementation to improve patient outcomes.

Our recommended definition of a chronic condition includes the previously recommended elements of chronicity, namely, duration, prognosis, pattern and producing consequences impacting individual's quality of life [16], with duration being the most important criteria as per the existing literature [54]. However, the exact duration is not universally established, with three most commonly mentioned intervals being three, six and twelve months [16]. Our consensus-defined duration of six months or more is aligned with the duration recommended by the World Organisation of Family Doctors and the Australian Institute of Health and Welfare [55].

Our study found the most commonly recommended cut-off for defining multimorbidity was three or more conditions based on both qualitative and quantitative Delphi round findings. Within the existing literature, authors have commonly used a threshold of either two or three for defining multimorbidity with the higher threshold being associated with lower estimated prevalence [56–59]. Our recommended threshold of three or more was higher than that recommended in the commonly known definitions, e.g., by World Health Organisation [13], European General Practice Research Network [15], and Agency for Healthcare Research and Quality [14]. From an epidemiological perspective, the implication of recommending a threshold of three in our setting would potentially result in estimating a lower multimorbidity prevalence. However, from a clinical perspective, adopting a higher threshold will result in a more discriminating definition and will help identify patients with higher care needs in local setting who will benefit from the holistic management [10].

We recommended a finalised list of 23 conditions which is higher than the 20 conditions recommended by Fortin et al. [47] Comparison with commonly known definitions, e.g., World Health Organisation [13], European General Practice Research Network [12], and Agency for Healthcare Research and Quality [55] on this parameter is not possible since none of these recommended a list of conditions for defining multimorbidity. In terms of the type of conditions included, few panellists in Delphi round 1 suggested adopting a more holistic

**Table 6. The consolidated findings for list of conditions across Delphi rounds 1 to 3.**

| S/N | Conditions | ICD-10 Codes | Original list | Round 1 feedback | Round 2 consensus voting & included in final list | Round 3 consensus voting & included in final list |
|---|---|---|---|---|---|---|
| 1 | Hyperlipidaemia | E78.5 (Hyperlipidaemia, unspecified) | ✓ | ✓ (Retained) | ✓ (Yes) | |
| 2 | Hypertension (high blood pressure) | I10 (Essential (primary) hypertension) | ✓ | ✓ (Retained) | ✓ (Yes) | |
| 3 | Diabetes (including pre-diabetes) | E09 (Impaired glucose regulation) | ✓ | ✓ (Retained) | ✓ (Yes) | |
| | | E099 (Impaired glucose regulation without complication) | | | | |
| | | E10.9 (Type 1 diabetes mellitus without complication) | | | | |
| | | E11.9 (Type 2 diabetes mellitus without complication) | | | | |
| | | E14.2 (Diabetes mellitus with incipient diabetic nephropathy) | | | | |
| | | E14.3 (Diabetes mellitus with retinopathy) | | | | |
| | | E14.31 (Unspecified diabetes mellitus with background retinopathy) | | | | |
| | | E14.64 (Unspecified diabetes mellitus with hypoglycaemia) | | | | |
| | | E14.73 (Unspecified diabetes mellitus with foot ulcer due to multiple causes) | | | | |
| 4 | Arthritis &/or rheumatoid arthritis | M06.99 (Rheumatoid arthritis, unspecified, site unspecified) | ✓ | ✓ (Retained) | ✓ (Yes) | |
| | | M15.9 (Osteoarthritis (OA)—Generalised) | | | | |
| | | M19.99 (Arthritis, unspecified, site unspecified) | | | | |
| 5 | Obesity | E66.9 (Obesity, unspecified) | ✓ | ✓ (Retained) | | ✓ (Yes) |
| 6 | Cardiovascular disease (angina, MI, AF, poor circulation of lower limbs) | I25.9 (Chronic ischaemic heart disease, unspecified) | ✓ | ✓ (Retained) | ✓ (Yes) | |
| | | I48 (Atrial fibrillation and flutter) | | | | |
| | | I70.20 (Atherosclerosis of arteries of extremities, unspecified) | | | | |
| | | I73.9 (Peripheral vascular disease, unspecified) | | | | |
| 7 | Asthma, COPD, or chronic bronchitis | J44.9 (Chronic Obstructive Pulmonary Disease, Unspecified) | ✓ | ✓ (Retained) | ✓ (Yes) | |
| | | J45.9 (Asthma, unspecified) | | | | |
| 8 | Chronic hepatitis | K76.9 (Liver disease, unspecified) | ✓ | ✓ (Retained) | | ✓ (Yes) |
| | | Z22.51 (Carrier of viral hepatitis B) | | | | |
| 9 | Stomach problem (reflux, heartburn, or gastric ulcer) | K21.9 (Gastro-oesophageal reflux disease without oesophagitis) | ✓ | ✓ (Retained) | | × (No) |
| | | K27.9 (Peptic ulcer, unspecified as acute or chronic, without haemorrhage or perforation) | | | | |
| 10 | Thyroid disorder | E03.9 (Hypothyroidism, unspecified) | ✓ | ✓ (Retained) | | ✓ (Yes) |
| | | E05.9 (Thyrotoxicosis, unspecified) | | | | |
| 11 | Stroke and TIA | G45.9 (Transient cerebral ischaemic attack, unspecified) | ✓ | ✓ (Retained) | ✓ (Yes) | |
| | | I64 (Stroke, not specified as haemorrhage or infarction) | | | | |

*(Continued)*

**Table 6.** (Continued)

| S/N | Conditions | ICD-10 Codes | Original list | Round 1 feedback | Round 2 consensus voting & included in final list | Round 3 consensus voting & included in final list |
|---|---|---|---|---|---|---|
| 12 | Heart failure (including valve problems or replacement) | I50.0 (Congestive heart failure) | ✓ | ✓ (Retained) | ✓ (Yes) | |
| | | I51.9 (Heart disease, unspecified) | | | | |
| 13 | Kidney disease or failure | N03.9 (Unspecified nephritic syndrome, unspecified) | ✓ | ✓ (Retained) | ✓ (Yes) | |
| | | N18.9 (Chronic kidney disease, unspecified) | | | | |
| 14 | Depression or anxiety | F32.20 (Severe depressive episode without psychotic symptoms, not specified as arising in the postnatal period) | ✓ | ✓ (Retained) | ✓ (Yes) | |
| | | F32.90 (Depressive episode, unspecified, not specified as arising in the postnatal period) | | | | |
| | | F41.1 (Anxiety disorder, unspecified) | | | | |
| 15 | Chronic urinary problem | N40 (Hyperplasia of prostate), | ✓ | ✓ (Retained) | | ✓ (Yes) |
| | | N39 (Other disorders of urinary system) | | ✓ (New addition) | | |
| | | N20.9 (Urinary calculus, unspecified) | | | | |
| | | Incontinence | | | | |
| 16 | Physical disability | H91.9 (Hearing loss, unspecified) | ✓ | × (Expanded into 2 categories of Functional Limitation and Cognitive Limitation) | Not Applicable | Not Applicable |
| | | Q79.9 (Congenital malformation of musculoskeletal system, unspecified) | | | | |
| 17 | Functional limitation | H91.9 (Hearing loss, unspecified) | | ✓ (Retained) | ✓ (Yes) | |
| | | Q79.9 (Congenital malformation of musculoskeletal system, unspecified) | | ✓ (Retained) | | |
| | | H26.9 (Cataract, unspecified) | | ✓ (New addition) | | |
| | | H54.9 (Unspecified visual impairment) | | ✓ (New addition) | | |
| | | Q89.9 (Congenital malformation, unspecified) | | ✓ (New addition) | | |
| | | Z89.4 (Acquired absence of foot and ankle) | | ✓ (New addition) | | |
| | | Z89.5 (Acquired absence of leg at or below knee) | | ✓ (New addition) | | |
| | | Z89.6 (Acquired absence of leg above knee) | | ✓ (New addition) | | |
| | | M67.99 (Disorder of synovium and tendon, unspecified) | | ✓ (New addition) | | |
| | | M79.89 (Other specified soft tissue disorders, site unspecified) | | ✓ (New addition) | | |
| | | Paraplegia | | ✓ (New addition) | | |
| | | Hemiplegia | | ✓ (New addition) | | |
| | | Enthesopathy | | ✓ (New addition) | | |
| 18 | Cognitive limitation | G80.9 (Cerebral palsy, unspecified) | | ✓ (New addition) | ✓ (Yes) | |
| | | Q90.9 (Down's syndrome, unspecified) | | ✓ (New addition) | | |
| | | F79.9 (Unspecified mental retardation without mention of impairment of behaviour) | | ✓ (New addition) | | |
| | | Autism | | ✓ (New addition) | | |
| | | ADHD | | ✓ (New addition) | | |

(*Continued*)

**Table 6.** (*Continued*)

| S/N | Conditions | ICD-10 Codes | Original list | Round 1 feedback | Round 2 consensus voting & included in final list | Round 3 consensus voting & included in final list |
|---|---|---|---|---|---|---|
| 19 | Any cancer in the last 5 years | C80 (Malignant neoplasm without specification of site) | ✓ | ✓ (Retained) | ✓ (Yes) | |
| 20 | Osteoporosis | M81.99 (Other osteoporosis, site unspecified) | ✓ | ✓ (Retained) | ✓ (Yes) | |
| 21 | Dementia or Alzheimer's disease | F03 (Unspecified dementia) | ✓ | ✓ (Retained) | ✓ (Yes) | |
| 22 | Colon problem (irritable bowel) | K58.9 (Irritable bowel syndrome without diarrhoea) | ✓ | ✓ (Retained) | | × (No) |
| 23 | Skin conditions | L70.9 (Other Acne) | | ✓ (New addition) | | × (No) |
| | | L20.8 (Other Atopic Dermatitis) | | | | |
| | | L40.0 (Psoriasis Vulgaris) | | | | |
| | | L40.8 (Other Psoriasis) | | | | |
| 24 | Chronic Pain | Pain | | ✓ (New addition) | | ✓ (Yes) |
| | | Chronic Fatigue | | | | |
| | | Fibromyalgia | | | | |
| 25 | Allergic rhinitis | J30.4 (Allergic rhinitis, unspecified) | | ✓ (New addition) | | × (No) |
| 26 | Gout | M10.9 (Gout, unspecified) | | ✓ (New addition) | | ✓ (Yes) |
| | | M10.99 (Gout, unspecified, site unspecified) | | | | |
| 27 | Other Mental Health Conditions | F20.9 (Schizophrenia, unspecified) | | ✓ (New addition) | ✓ (Yes) | |
| | | F22.9 (Delusional disorder) | | | | |
| | | F29 (Unspecified nonorganic psychosis) | | | | |
| | | F31.9 (Bipolar affective disorder, unspecified) | | | | |
| | | F48.9 (Neurotic disorder) | | | | |
| | | F55.9 (Unspecified harmful use of non-dependence producing substance) | | | | |
| | | F99 (Mental disorder, not otherwise specified) | | | | |
| | | G47.0 (Disorders of initiating and maintaining sleep [insomnias]) | | | | |
| | | Z86.5 (Personal history of other mental and behavioural disorders) | | | | |
| | | PTSD | | | | |
| | | OCD | | | | |
| | | Chronic narcotic dependency syndrome/drug abuse | | | | |
| | | Personality disorder | | | | |
| | | Phobia | | | | |
| | | Somatoform disorders/somatic symptom disorder | | | | |
| | | Eating disorders | | | | |
| | | Alcohol abuse | | | | |
| | | Burnout | | | | |

(*Continued*)

**Table 6.** (Continued)

| S/N | Conditions | ICD-10 Codes | Original list | Round 1 feedback | Round 2 consensus voting & included in final list | Round 3 consensus voting & included in final list |
|---|---|---|---|---|---|---|
| 28 | Neurological Disorders | G40.90 Epilepsy, unspecified, without mention of intractable epilepsy |  | ✓ (New addition) | ✓ (Yes) |  |
|  |  | G20 Parkinson's disease |  |  |  |  |

Note: Under Original List column: "✓" indicates the condition was present in the original list; Under Round 1 feedback column: "✓ (Retained)" indicates the condition was retained from the original list based on feedback from Delphi panellists, "✓ (New addition)" indicates the condition was newly added based on feedback from Delphi panellists; Under Round 2 consensus voting & included in the final list column: "✓ (Yes)" indicates that the condition gained consensus to be included in the final list based on voting by Delphi panellists in Round 2; Under Round 3 consensus voting & included in final list column: "✓ (Yes)" indicates that the condition gained consensus to be included in the final list based on voting by Delphi panellists in Round 3, "✗ (No)" indicates that the condition did not gain consensus to be included in the final list based on voting by Delphi panellists in Round 3.

biopsychosocial perspective. While this was incorporated in defining a chronic condition, our definition of multimorbidity did not explicitly include the biopsychosocial component. Across the numerous existing definitions, some like the European General Practice Research Network have advocated for a definition inclusive of the biopsychosocial component [60], while others have suggested that simpler definitions are more acceptable [61].

Our study findings generally align with the international literature on the types of conditions included in the finalised list with inclusion of commonly reported conditions like hypertension, arthritis, diabetes and cardiac problems [62]. While our definition of multimorbidity includes chronic health conditions similar to definitions by the World Health Organisation and the Agency for Healthcare Research and Quality [14, 15], it does not include acute conditions as recommended by the European General Practice Research Network definition of multimorbidity [15]. This exclusion is apt within the Singaporean context of a developed country, with a rapidly aging population [63] characterised by transition away from communicable to chronic conditions, which accumulate with increasing age [64]. Hence, from an epidemiological perspective, exclusion of acute conditions should not significantly impact the multimorbidity prevalence estimates generated in our local setting based on our recommended definition. Moreover, the exclusion of acute conditions was based on consensus voting by the Delphi panellists, in line with including conditions with the largest contribution to the disease burden within Singapore like cardiovascular diseases, cancer, mental disorders etc. [65]. From a clinical perspective, the acute complaints usually seen in primary care are generally less severe and may not significantly contribute towards the multimorbidity burden [56].

While there were 23 types of conditions included in the finalised list, three were excluded due to lack of consensus, namely, stomach, colon and skin conditions. Panellists qualitatively shared they perceived both stomach and skin conditions to be benign and not having a significant impact. For colon problems, panellists additionally shared the difficulty in diagnosing, which may result in inaccurate diagnosis and prevalence estimation, and perception of colon problems as more of a lifestyle-related condition. In agreement with our findings, a systematic review from South Asia reported less than half of the studies included stomach or skin conditions and none had colon problems [62]. Additionally, the standardised prevalence ratios for these conditions in the Singaporean primary care setting were reported to be relatively lower (2.52%, 0.20% and 0.08% for stomach problems, colon problems and skin conditions, respectively) as compared to other chronic conditions [19].

Although chronic pain was included in the final list, currently there are no standardised, recommended guiding principles for documenting chronic pain in the public primary care

coding system in Singapore, which primarily uses ICD-10 coding system. This unavailability of systematic guidance may result in inconsistent coding using ill-defined symptom based codes [66]. While one previous study reported being unable to map available ICD-10 codes to the category of '*chronic musculoskeletal condition causing pain or limitation*' [19], only one [67] out of the remaining studies measuring multimorbidity in Singapore included chronic pain documented via self-reports [67]. As one of the practical implications of this work, we would be engaging the relevant stakeholders in the primary care sector to derive mutually agreed upon coding principles for chronic pain.

From the perspective of multimorbidity frameworks, Willadsen et al. [68] recommended including three categories of diseases, risk factors and symptoms for describing multimorbidity. While the conditions suggested by our Delphi panellists included these three categories, we did not explicitly categorise the conditions into diseases, risk factors and symptoms. However, comparing our finalised list with the most frequently reported diseases, risk factors and symptoms, our finalised list comprises diseases like diabetes, cardiovascular diseases, stroke etc., risk factors like hypertension, hyperlipidaemia, obesity etc., and symptoms like chronic pain, hearing loss, incontinence etc. While from an epidemiological perspective, including risk factors and symptoms will result in potentially higher multimorbidity prevalence [69], from a clinical perspective, inclusion of risk factors and symptoms will result in incorporation of patients' views for more holistic management of patients with multimorbidity.

Focussing on local literature, our consensus-based cut-off of three or more conditions for defining multimorbidity was higher than the cut-off of two or more used previously [5, 8, 35–37, 67]. Comparing the total number of conditions included, four [5, 8, 36, 67] out of the five studies from Singapore had the total number of conditions in their list lower than the 23 conditions included in our finalised list. Moreover, for three of these studies [8, 35, 36], the source of the list was not mentioned, which makes it challenging to assess the validity. When comparing the type of conditions included across these studies, our finalised list of conditions included all conditions from three of the above four studies [8, 35, 36, 67]. For the study by Quah and colleagues, we included all but one condition of gastrointestinal diseases based on feedback from the Delphi panellists [5].

From an epidemiological perspective, since looking at individual dimensions for defining multimorbidity may result in varying prevalence findings, it is important to understand that the definition of multimorbidity is multi-dimensional and hence, it is not plausible to exactly know the impact of a recommended definition on prevalence of multimorbidity when compared with others. It is more meaningful to focus on the clinical relevance of such definitions along with understanding the methodological accuracy. From a clinical perspective, adoption of a higher cut-off in our definition is more meaningful since it would result in identification of patients with higher care needs, who would benefit from the holistic management in a primary care setting. It is not feasible to assess the accuracy of the local studies as none of them explained in detail the science behind developing their definition. In contrast, our recommended definition of multimorbidity, including the list of 23 conditions, was a result of consensus-voting by a group of Delphi panellists, who had both expertise in the field of multimorbidity and practical insights into managing patients with multiple chronic conditions in a primary care setting. Therefore, our definition is potentially more suited for defining and measuring multimorbidity within the Singapore primary care setting. In the absence of a universally recommended definition of multimorbidity and lack of consensus, our chosen Delphi methodology is one of the most appropriate methods to gain consensus. Moreover, the conduct and reporting of our Delphi study was done in accordance with CREDES [40], which illustrates the robustness of our study and the validity of our findings. Throughout the study, we upheld the core tenets of the Delphi process, i.e., maintaining quasi-anonymity, practising

iteration and controlled feedback, responses of the group being analysed in a statistical manner and compared with pre-defined consensus thresholds and sharing of results with Delphi panellists after each round. All of the above led to the final recommendation of a consensus-derived, context-based, clinically relevant definition of multimorbidity, which was rooted in sound methodology and well-supported by our study findings [68, 69].

While existing literature highlights heterogeneity in prevalence estimates based on the type of data source used [70–73], there is no clear consensus on which data source to use. Hence, our findings related to types of data to use will provide a good starting point in the local context to choose an appropriate data source. For example, if researchers are studying multimorbidity from patients' perspective (e.g., impact on multimorbidity on quality of life, self-rated health or treatment burden etc.), they may want to include Patient Self-Reported Outcomes as the data source. Similarly, Electronic Health Records may be the source from provider's perspective (e.g., association of multimorbidity with consult time). Lastly, Ministry of Health administrative data may be the source from health systems' perspective (e.g., quantifying the healthcare utilisation associated with multimorbidity).

The main strengths of our study were having both public and private primary care representation in the Delphi panel and a good response and retention rate across all Delphi rounds. Additionally, we conformed to the recommended reporting standards as per the CREDES guidelines [40]. Maintaining quasi-anonymity enabled participants not to be aware of each other's responses, thus limiting the pressure to conform to convergence which could threaten the study validity [74]. Collecting multiple rounds of responses ensured concurrent validity. To the best of our knowledge, we are the first to gain consensus on this relevant and debated topic of defining and measuring multimorbidity within the ambulatory primary care setting of Singapore.

Our study has several limitations. Certain components and chronic conditions did not reach a consensus rating at the end of Delphi round 3. As mentioned above, our finalised list did not include stomach, colon and skin conditions which is similar to other local findings [5, 8, 35–37, 67]. Moreover, the locally reported standardised prevalence for these conditions is relatively lower as compared to other conditions within the finalised list [19]. Our approach of conducting three Delphi rounds is in accordance with the Delphi methodology recommendations, considering the principle of diminishing returns, potential participant fatigue and steep drop-out rates with more rounds [75, 76]. Since our study primarily aimed to gain consensus within Singapore's primary care setting, the generalisability of our findings would be limited. Within Singapore, the generalisability of our findings may be potentially affected by our sample including mainly male and younger panellists. Another limitation inherent in the Delphi technique is that panellists do not get the opportunity for elaborating their shared views. However, this should have minimal impact on our findings considering consensus was sought on clearly defined scope of work.

On the practice front, our findings will inform the conceptualisation of multimorbidity at the national level in Singapore, potentially standardising the measurement of multimorbidity within our ambulatory primary care setting. This would subsequently facilitate planning and resource allocation for patients with multimorbidity in Singapore. Having defined multimorbidity, next we would estimate the national prevalence of multimorbidity using this agreed-upon definition and analysing its impact on healthcare utilisation and costs.

## Conclusion

With the aim to conduct a Delphi study to gain consensus on the definition, conceptualisation of multimorbidity and the list of chronic conditions used to categorise patients with

multimorbidity in the ambulatory primary care setting of Singapore, we reported the consensus-derived definition of a chronic condition, multimorbidity, recommended data sources from multiple perspectives and the finalised list of 23 conditions (inclusive of diseases, risk factors and symptoms) for measuring multimorbidity. For a condition to be chronic, it should last for six months or more, be recurrent or persistent, impact patients across multiple domains and require long-term management. The consensus-derived definition of multimorbidity is the presence of three or more chronic conditions from a finalised list of 23 chronic conditions. Our findings will inform multimorbidity conceptualisation at the national level in Singapore, standardise multimorbidity measurement in primary care and facilitate resource allocation. On the research front, we would estimate the national prevalence of multimorbidity using this agreed-upon definition and analyse its impact on healthcare utilisation and costs.

## Supporting information

**S1 Appendix. CREDES checklist.**
(DOCX)

**S2 Appendix. Delphi round 1 survey.**
(DOCX)

**S3 Appendix. Delphi round 2 survey.**
(DOCX)

**S4 Appendix. Delphi round 3 survey.**
(DOCX)

**S5 Appendix. Finalised list of chronic conditions for defining multimorbidity.**
(DOCX)

## Acknowledgments

We would like to thank all the participants in our study for their participation and cooperation.

## Author Contributions

**Conceptualization:** Shilpa Tyagi, Gerald Choon-Huat Koh, Eng Sing Lee.

**Data curation:** Shilpa Tyagi.

**Formal analysis:** Shilpa Tyagi, Victoria Koh, Gerald Choon-Huat Koh, Lian Leng Low, Eng Sing Lee.

**Funding acquisition:** Gerald Choon-Huat Koh, Eng Sing Lee.

**Investigation:** Shilpa Tyagi, Victoria Koh.

**Methodology:** Shilpa Tyagi, Victoria Koh, Gerald Choon-Huat Koh, Lian Leng Low, Eng Sing Lee.

**Writing – original draft:** Shilpa Tyagi.

**Writing – review & editing:** Shilpa Tyagi, Victoria Koh, Gerald Choon-Huat Koh, Lian Leng Low, Eng Sing Lee.

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
