## [Decision Letter · Decision Letter 0]

2 Nov 2021

PONE-D-21-28758Defining and measuring multimorbidity in primary care in Singapore: results of an online Delphi studyPLOS ONE

Dear Dr. Koh,

Thank you for submitting your manuscript to PLOS ONE. After careful consideration, we feel that it has merit but does not fully meet PLOS ONE’s publication criteria as it currently stands. Therefore, we invite you to submit a revised version of the manuscript that addresses the points raised during the review process.

For further considering your manuscrito at PLOS ONÉSIMO, please see and address the comments and requests by two independent reviewers below.

We look forward to receiving your revised manuscript.

Kind regards,

Ferrán Catalá-López

Academic Editor

PLOS ONE

3. PLOS requires an ORCID iD for the corresponding author in Editorial Manager on papers submitted after December 6th, 2016. Please ensure that you have an ORCID iD and that it is validated in Editorial Manager. To do this, go to ‘Update my Information’ (in the upper left-hand corner of the main menu), and click on the Fetch/Validate link next to the ORCID field. This will take you to the ORCID site and allow you to create a new iD or authenticate a pre-existing iD in Editorial Manager. Please see the following video for instructions on linking an ORCID iD to your Editorial Manager account: https://www.youtube.com/watch?v=_xcclfuvtxQ.

Reviewers' comments:

Reviewer's Responses to Questions

**Comments to the Author**

1. Is the manuscript technically sound, and do the data support the conclusions?

Reviewer #1: Partly

Reviewer #2: Yes

2. Has the statistical analysis been performed appropriately and rigorously? 

Reviewer #1: Yes

Reviewer #2: Yes

3. Have the authors made all data underlying the findings in their manuscript fully available?

Reviewer #1: Yes

Reviewer #2: Yes

4. Is the manuscript presented in an intelligible fashion and written in standard English?

Reviewer #1: Yes

Reviewer #2: Yes

5. Review Comments to the Author

Reviewer #1: This manuscript covers a current and valuable topic in chronic condition care: what is meant by multimorbidity? The key reasons for defining multimorbidity is to facilitate understanding the prevalence of multimorbidity, the health impact and provide targets for clinical improvement and research studies. This study aims to define multimorbidity for a primary care audience in Singapore.

The authors’ rationale for the narrow focus of their multimorbidity definition is not well-explained in the manuscript nor why the authors think a new definition from primary care in Singapore is needed compared to adapting a current definition. The manuscript should explain why the authors think their definition is needed and what the implications for its use are. This should be better articulated in the intro and discussion. The intro explains why a universal definition is needed, but the paper doesn’t offer that or is this a first step towards that goal?

The paper would be stronger if it more clearly compared their new definition against existing definitions (3 vs 2 conditions, out of a list of 27, and which conditions included) and what these means clinically. Would some patients have multimorbidity form the new definition vs not another, and does this matter? Consider adding more definitions – US AHRQ definition is missing. I saw WHO, European and Australia mentioned but the comparisons were limited. How might it change clinical care to have a 3 vs 2 cut off, to have a different list of conditions or different subgroupings of conditions?

The paper would also be stronger if the authors explained why specific conditions were included and why they were grouped in the first round. Why were depression and anxiety grouped together but bipolar was not included with them and rather with “other”? Why was asthma with COPD and not separate? Some of the groupings are for very similar conditions but others could be argued as separate chronic conditions – the rationale for these groupings pre-Delphi needs to be made clear. Also, were inflammatory bowel diseases included? I only saw IBS. What was the rationale for the limited conditions included at the beginning? Please explain how these were chosen. Also, address more explicitly “risk factors” vs conditions (how were obesity, hyperlipidemia, etc handled)

Also, if they explained why specific conditions are needed for a definition of multimorbidity when they also give a cut off for overall number of conditions and a definition of chronic condition.

The paper is lacking a limitations section. Only one limitation is mentioned within a strengths section. One major limitation is that the conditions are those considered important for a Singaporean population. I think the answers on which conditions matter for multimorbidity would be different in a different population (IBS and GERD might be important in the US, for instance) and that needs to be stated. Also, are there any limitations to the sample? The sample seemed young and mostly male.

For Table 5, I recommend putting a column with a check to mark that the condition was included in the final list of conditions. This would improve reading clarity and this important result should be in a main table,. Not just the Appendix.

This is a valuable topic and the paper would add more value to the literature with more clearly stated rationale for needing a Singaporean primary care-based definition (implications for use) and comparisons to current definitions including differences in clinical implications.

Reviewer #2: Thank you for this interesting article. Multimorbidity is an increasingly important topic. In this respect, this work represents a valuable contribution in this field. The article is very well structured, the goal is clearly described and the procedure is presented transparently. On the positive side, you used the CREDES checklist to report the results.

Still, I noticed a few things:

Abstract: p. 2 (row 31-32) The increasing response rate is irritating. Even if it is explained later from the flowchart. Maybe the specification like this: 61.7% (37/40), 86.5% (32/37) and 93.8% (30/32)?

p.4 (row 71-72): What exactly do you mean by "to improve outcomes of patients with multimorbidity?" Please explain in more detail.

p.3 (row 47): In my opinion, source 4 does not show the named spread (“it ist common….”) of multimorbidity.

p.7 (row159-160): You gave the participants literature so that they would receive unbiased and new ideas. But aren't the ideas then influenced by the literature? Or have I misunderstood that?

p.9 (row179): “the most common elements” is somewhat unspecific. How many times was it mentioned?

p.9 (row 185): “Most panallists” is also unspecific. How many?

p. 16: Please adjust the size of table 4 to all other tables.

p.26: (row429-430): “enabled participants not to be aware of each other’s responses,” But the results of the previous rounds were reflected in them, right?

Discussion: Please detail to the point "Measuring the burden of multimorbidity". This was raised in Delphi and shown for all rounds. However, it does not appear in the discussion or in the conclusion.

The flowchart (Fig. 1) is very blurry. Perhaps it would be better not to insert it as a picture?

Tab. 5 extends over 3 pages. Is it possible to put this on one page?

6. PLOS authors have the option to publish the peer review history of their article (what does this mean?). If published, this will include your full peer review and any attached files.

Reviewer #1: No

Reviewer #2: No

---

## [Author Response · Author response to Decision Letter 0]

17 Dec 2021

We have addressed all the comments shared by the reviewers and our detailed point-by-point responses are included in the uploaded 'Response to Reviewers' document. Please refer to this document for further details. Thank you.

---

## [Decision Letter · Decision Letter 1]

27 Jan 2022

PONE-D-21-28758R1Defining and measuring multimorbidity in primary care in Singapore: results of an online Delphi studyPLOS ONE

Dear Dr. Koh,

Thank you for submitting your manuscript to PLOS ONE. After careful consideration, we feel that it has merit but does not fully meet PLOS ONE’s publication criteria as it currently stands. Therefore, we invite you to submit a revised version of the manuscript that addresses the points raised during the review process.

The manuscript has been assessed by two reviewers, and their comments are appended below. One of the reviewers have raised major concerns regarding the analyses reported, the design of the study, the statistical analyses and the validity of the results reported. Please, could you carefully revise the manuscript to address the concerns raised?

We look forward to receiving your revised manuscript.

Kind regards,

Elisa Panada

Associate Editor

PLOS ONE

Reviewers' comments:

Reviewer's Responses to Questions

**Comments to the Author**

1. If the authors have adequately addressed your comments raised in a previous round of review and you feel that this manuscript is now acceptable for publication, you may indicate that here to bypass the “Comments to the Author” section, enter your conflict of interest statement in the “Confidential to Editor” section, and submit your "Accept" recommendation.

Reviewer #1: (No Response)

Reviewer #2: All comments have been addressed

2. Is the manuscript technically sound, and do the data support the conclusions?

Reviewer #1: No

Reviewer #2: Yes

3. Has the statistical analysis been performed appropriately and rigorously? 

Reviewer #1: I Don't Know

Reviewer #2: Yes

4. Have the authors made all data underlying the findings in their manuscript fully available?

Reviewer #1: No

Reviewer #2: Yes

5. Is the manuscript presented in an intelligible fashion and written in standard English?

Reviewer #1: No

Reviewer #2: Yes

6. Review Comments to the Author

Reviewer #1: I appreciate the opportunity to review this paper again and appreciate the authors’ attempts to address review concerns. Unfortunately, the authors have made minimal changes to address the concerns brought forth by the reviewers. Many of the changes are surface level and not well-integrated into the paper and bring new issues to light. This paper remains potentially important however the importance of the study is in doubt as it is not well-justified in the Introduction or well-supported by the Methods and Results used to address the gap in the literature in a meaningful, accurate way.

The Methods and Results remain unclear. The authors also do not explain well how this new definition makes a helpful impact on the field. They merely explain how the definition is different than others but not why it is better, more accurate or more meaningful. The lack of clarity in the Methods, Results and Discussion make it difficult to determine the scientific validity of the work. Based on the manuscript as written, I do not see how readers would be able to use the authors’ new multimorbidity definition or why it would be chosen over an established definition. More complete rewrites are needed to address these methodological and scientific meaning concerns.

Introduction

-says that definitions are without much convergence and then lists 3 definitions that are the same: 2 or more conditions or health factors. The European GP Research Network could be considered different in that it names acute conditions or risk factors as being included along with a single chronic condition, rather than requiring 2 or more chronic conditions (which include risks factors), but these are not that different. More depth is needed here as well as more definitions that are cited later in the paper.

-The intro needs to demonstrate where there are meaningful differences in definitions and why the authors’ new definition is needed. The Singaporean-only definition for primary care does not help with consensus across multiple countries or for comparison between studies outside PC in Singapore. So why is this definition needed? What does it give you? Why can’t a current definition that has already been developed and tested be used? Is there something unique in Singapore that requires its own definition? This must be justified more. It seems to me that per-country definitions will cause more fragmentation rather than clear comparisons between countries and regions, so you need to well-justify the need for country-specific multimorbidity definitions. (Especially as these definitions seem to be based on current prevalence patterns in the population that can change based on the population’s health and the doctor’s diagnostic patterns - so how often should these definitions be revisited?)

-The reference 16 describes multiple measures being compared but that methodological detail is not in this manuscript, nor is an explanation of why the range in prevalence found in #16 is a clinical concern (likely it is, but that should be supported in this manuscript.)

-If #16 found that Fortin’s list is the best and prediabetes and physical diabetes should be added, why is this Delphi study needed? What does the study add to what was found in #16? That justification would help.

-the authors revisions to justify why PC are incomplete. More clear justification is needed on why a Singapore-only, PC-only definition is needed compared to using a current definition. Why are current definitions more faulty for Singaporean PC than for other populations? Multimorbidity definitions do lack some consensus and do have their faults, but why does there need to be on for Singapore primary care particularly? Do the authors argue that each country should have their own PC definition?

-How will having this be stakeholder driven will help develop interventions and tracking programs. The ideas are here but need clarity and specifics from the literature.

-“there still remains heterogeneity in both the overall methodology and the list of chronic conditions considered, highlighting the need to develop a new, consensus-derived definition of multimorbidity within the primary care setting in Singapore.” This needs specific justification and citations. Is this referring to the work in #16? Those details need to be above (don’t make the reader go to the cited paper in order to understand your justification) and then restating in summary here.

Methods:

-“The list proposed 192 by Fortin et al. [37] was modified to make it suitable for use in primary care setting in Singapore 193 with inclusion of pre-diabetes under the chronic condition of ‘diabetes’ and also inclusion of 194 ‘physical disability’ in the modified list.” This section needs to also explain that the work in 16 found that this definition had the most accurate prevalence (if true) and how the authors concluded that prediabetes and physical disability were needed.

Results

-the authors did not explain why bipolar disorder was grouped separately from anxiety and depression. They seem to have made that choice separately from what the Delphi round participants suggested. The authors seemed to have started with the idea of Fortin + prediabetes and physical disability and ended there. It is not clear if they used any data from Delphi to adjust their definition.

-how was the impact of the chronic condition (Table 3) used in the final definition of multmorbidity? That seems to be included as a separate entity and not incorporated.

-Why does Table 3 have empty columns for Round 3? That is odd. These should be removed and a comment added to the legend that these items were not considered din round 3. Otherwise, fill in the table for Round 3 with “not considered.”

-For the cut off of conditions, how many panelists agreed to 2 or more conditions (the current most common count per your introduction). How many to 1 condition plus an acute condition or other risk factor? (Per the Euro GP Research Network)? It seems the authors are only presenting the findings that support their final conclusions but the readers will want to see the other relevant data.

-The conditions in S3 and S4 need to be listed in the main manuscript, not buried in a survey in the appendix.

-The results from each ound are unnecessarily wordy and hard to follow. A terse re-write would benefit the reader.

- The authors did not address when IBD is left out.

-why is gout separate from OA or RA?

-It is very hard to follow which conditions were provided to the panelists in Round 1 and which they suggested themselves, then how these were presented in Round 2 and Round 3 to get to your consensus. It is also hard to see how the list developed through this work compares to what fortis et al (or others) suggest. A clear flow diagram and table would help. Without clarity in these methods and results, we cannot see the scientific validity of the work.

Discussion:

-The first paragraph should provide your concise definition (your study findings) and possibly suggested use of the definition.

-The discussion next seems to present new qualitative findings and methods. All results and methods should be in their sections. The discussion should serve to place these findings in the current literature and give implications.

-the focus in the discussion on 2 vs 3 as a cut off needs to be set up in the intro and be well-supported by your research methodology and results. That is ignored in the paper.

-What does this mean:” The inclusion of acute conditions for defining multimorbidity may be more relevant in developing settings where the prevalence of such conditions is higher as compared to a developed setting like Singapore. [47]” Europe is a developed setting. Is the European GP Research Network actually in developing settings?

-the authors have made helpful additions to the discussion based on reviewer feedback but this is not well integrated into the discussion resulting in a disjointed discussion. The discussion should be re-written to fully incorporate the comparison with current literature and implications on the authors proposed new definition.

-“Comparing our finalised list of conditions with existing local literature, 4four [5, 8, 26, 468 50] out of the 5five studies from Singapore had the total number of conditions in their list lower than the 23 conditions included in our finalised list. Hence, the multimorbidity prevalence estimates generated by our list are expected to be higher as compared to most of previous studies from Singapore. When comparing the type of conditions included across these studies, our finalised list of conditions included all conditions from three of the above four studies. [8, 25, 26, 50] For the study by Quah and colleagues, we included all but one condition of gastrointestinal diseases based on feedback from the Delphi panellists. [5]” More is needed here. What does it mean if your multimorbidity estimates are higher? Of course they will be higher, more conditions will be included, but is that correct? Are your conditions more accurate because they were brought in by the Delphi methods? What were the definitions used in the other papers and why are they inadequate?

-In the Discussion, it is not enough to simply list that you have a higher cut off and more conditions, so you will have both a higher threshold to identify multi-morbidity (3+) and a lower one (more conditions). You needed to explain the science behind why your definition is more accurate or clinically relevant, so it is more correct than the previous definitions. Being different is not enough. The readers want to hear how your scientific method was better, resulting in a more accurate and more useful definition.

-“Willadsen et al. [17] summariszed …..” You need to comment on if your Delphi panel thought risk factors and symptoms were needed. Also, state who decided to call “chronic pain” a symptom and not a condition (state if the 3 lists you give are from #17 or another source.)

-Again in the above paragraph you state that your measure will identify more patients with multimorbidity but do not mention if that is more accurate or what it means for clinical care.

-In the discussion, the definition of chronic condition and rationale around stomach problems seem to contradict with including symptoms (pain) or risk factors (asymptomatic hypertension). This needs to be more clearly explained.

-“However, currently, there are no standardised, recommended codes for 531 documenting chronic pain in the public primary care coding system in Singapore. While one 532 previous study reported being unable to map available ICD-10 codes to the category of ‘chronic 533 musculoskeletal condition causing pain or limitation’, [16] only one out of the remaining 534 previous studies measuring multimorbidity in Singapore included chronic pain in their list of 535 conditions.” How is there no way to diagnose chronic pain in Singapore? There are numerous ICD-10 codes for chronic pain. This needs more explanation. Also the “only one of the remaining studies” needs a citation. This sentence is hard to follow.

-“Hence, our findings will provide a good starting point in the local context to 547 choose an appropriate data source based on different perspectives.” How do you suggest suing these 3 different data sources? Choose one? Add them together? Average?

-The Discussion overall lacks depth and seems to not understand why an accurate definition of multimorbidity is needed or what it means. The authors contradict themselves, supporting the benefits of a stricter definition of multimorbidity with 3+conditions to have an underestimate of multimorbiity such that those identified are the sickest. But then the authors say that they include more conditions (including prediabetes, any kidney disease and hypertension which can all be quite mild from a patient-perspective) makes them able to identify more conditions. This is a contradiction. The authors also only compare the counts of included conditions compared to other definitions of multimorbidity but don’t explain the meaning between them and why their new definition might better serve their audience clinically. There is some mention of this around stomach issues being less common in Singapore. More of this depth of discussion is needed and should be brought together into a clear, convincing argument.

-The discussion is quite long and should be tightened on rewrite.

Conclusion:

-the authors state: “The 593 consensus-derived definition of multimorbidity is the presence of three or more chronic 594 conditions from a finalised list of 23 chronic conditions” however they also included what they call Risk Factors and Symptoms, and do not state that in their definition.

Tables

-see comment on Table 3 above

-The authors claim Table 5 cannot fit onto 1 page but it clearly can fit onto 2 with some simple editing. Even though the journal allows multiple pages, I encourage the authors to take into consideration readability of the paper for their audience and take the reviewer’s suggestion to make this table more readable.

Reviewer #2: Thank you so much for the changes and further explanations made in accordance with the comments.

No further comments from my side.

7. PLOS authors have the option to publish the peer review history of their article (what does this mean?). If published, this will include your full peer review and any attached files.

Reviewer #1: No

Reviewer #2: No

---

## [Author Response · Author response to Decision Letter 1]

10 Mar 2022

Please refer to the uploaded 'Response to Reviewers' document for our detailed responses to all the comments given by editor and reviewers. Thank you.

---

## [Decision Letter · Decision Letter 2]

6 Jul 2022

PONE-D-21-28758R2Defining and measuring multimorbidity in primary care in Singapore: results of an online Delphi studyPLOS ONE

Dear Dr. Koh,

Thank you for submitting your manuscript to PLOS ONE. After careful consideration, we feel that it has merit but does not fully meet PLOS ONE’s publication criteria as it currently stands. Therefore, we invite you to submit a revised version of the manuscript that addresses the points raised during the review process.

Please note the comments from Reviewer #1 below. The reviewer has noted in their report that they were not able to access some of the review files; I therefore contacted them with the missing documents, and they have now also been able to assess the changes you have made in full. As such, in addition to the comments in their report copied below, please also provide a response to the reviewer's follow-up comments: "I have reviewed the author’s response to reviewers. They have many changes successfully. However, for other recommended clarifications, they have added long explanations that are vague and do not get to the heart of the issue that I raised. Some concerns were not addressed or only partially addressed. More concise, terse changes would have benefitted the article.

While this is interesting work, and seems to have been done well, I still have trouble with the underlying premise that a single country, single practice setting definition of multimorbidity is needed or would be beneficial. I could see this doing harm to the clinical and research communities by causing division and increased heterogeneity in measures. The authors rationale is not clearly explained, despite my comment to them that it should be supported more clearly, and their rationale is at times contradictory." Please address all of the reviewer's comments when revising your manuscript. In particular, please ensure you respond to their concerns regarding increased heterogeneity in measures, and regarding clarity of the rationale for this study.

We look forward to receiving your revised manuscript.

Kind regards,

Hugh Cowley

Senior Editor

PLOS ONE

Journal Requirements:

Reviewers' comments:

Reviewer's Responses to Questions

**Comments to the Author**

1. If the authors have adequately addressed your comments raised in a previous round of review and you feel that this manuscript is now acceptable for publication, you may indicate that here to bypass the “Comments to the Author” section, enter your conflict of interest statement in the “Confidential to Editor” section, and submit your "Accept" recommendation.

Reviewer #1: All comments have been addressed

2. Is the manuscript technically sound, and do the data support the conclusions?

Reviewer #1: Yes

3. Has the statistical analysis been performed appropriately and rigorously? 

Reviewer #1: Yes

4. Have the authors made all data underlying the findings in their manuscript fully available?

Reviewer #1: Yes

5. Is the manuscript presented in an intelligible fashion and written in standard English?

Reviewer #1: Yes

6. Review Comments to the Author

Reviewer #1: I appreciate the opportunity to review this paper again and appreciate the authors’ attempts to address review concerns. The paper is much clearer and more complete. The methods and results are now clear. The manuscript fills an important gap in the literature for Singapore primary care and perhaps beyond. However, from the authors efforts to incorporate the reviewer’s comments, the Introduction and Discussion are now much too long for a standard journal article. I was also unable to find the Response to Reviewers comments within the online reviewer platform.

The Tables are much improved and add to the understanding of the methods and results.

Please revise the Introduction and Discussion to be shorter. The content is good but a terse re-write is needed.

A few additional comments:

-The paper needs to be consistent about the multimorbidity measure being for Singapore with the potential to be used broadly. Often it come across that this is a general use measure and the study design does not support that.

-Introduction: “While the World Health Organisation has defined multimorbidity as “being affected by two or more chronic health conditions”,[13] the European General Practice Research Network 53adopted a more comprehensive approach and defined multimorbidity as “any combination of 54chronic disease with at least one other disease (acute or chronic) or bio-psychosocial factor 55(associated or not) or somatic risk factor.” [14] Agency for Healthcare Research and Quality defines multiple chronic conditions as presence of “two or more chronic physical or mental health conditions.”” Please check the gramma of the first sentence and incorporate the 2nd sentence with the first.

-Introduction: The authors state that previous measure do not define what a chronic condition is and that is false. The authors themselves later explain how chronic conditions are defined by various groups.

-Discussion: “Our study found the most commonly reported cut-off for defining multimorbidity in 477qualitative Delphi Round 1 was 3 or more conditions.” Would be better revised as “Our study found the most commonly recommended cut-off for defining multimorbidity in qualitative Delphi Round 1 was 3 or more conditions”

-Discussion: Your argument that pain cannot be mapped to ICD-10 codes is superficial. There are certainly chronic pain codes in ICD-10 (G89.4 for one) and attempts have been made to list all pain codes. It does remain a challenge and I encourage you to rewrite this section around the challenge rather than stating there are not pain codes in ICD-10.

7. PLOS authors have the option to publish the peer review history of their article (what does this mean?). If published, this will include your full peer review and any attached files.

Reviewer #1: No

---

## [Author Response · Author response to Decision Letter 2]

20 Aug 2022

Please refer to the Response to Reviewer document for our detailed responses to the comments kindly given by the editor and the reviewer. Thank you very much.

---

## [Editor Report · Decision Letter 3]

21 Nov 2022

Defining and measuring multimorbidity in primary care in Singapore: results of an online Delphi study

PONE-D-21-28758R3

Dear Dr. Koh,

We’re pleased to inform you that your manuscript has been judged scientifically suitable for publication and will be formally accepted for publication once it meets all outstanding technical requirements.

Kind regards,

Kelvin I. Afrashtehfar, M.Sc., D.D.S.,Dr. med. dent., FRCDC

Academic Editor

PLOS ONE

Additional Editor Comments (optional):

Thank you for your efforts and for considering PLOS ONE.
---

## [Editor Report · Acceptance letter]

23 Nov 2022

PONE-D-21-28758R3 

Defining and measuring multimorbidity in primary care in Singapore: results of an online Delphi study 

Dear Dr. Koh:

I'm pleased to inform you that your manuscript has been deemed suitable for publication in PLOS ONE. Congratulations! Your manuscript is now with our production department. 

Kind regards, 

on behalf of

Dr. Kelvin I. Afrashtehfar 

Academic Editor

PLOS ONE